Surprising diversity of new plasmids in bacteria isolated from hemorrhoid patients

Wang Yihua 1
Su Wenya 2
Zeng Xiang 3
Liu Zhaopeng 1
Zhu Jiaming 4
Wang Mingyu 2
Li Ling 2 lingli@sdu.edu.cn
Shen Wenlong 1 swl028073@qlyyqd.com
1 Department of Anorectal Surgery, Qilu Hospital (Qingdao), Cheeloo College of Medicine, Shandong University , Qingdao , China
2 State Key Laboratory of Microbial Technology, Microbial Technology Institute, Shandong University , Qingdao , China
3 Department of Anorectal Surgery, Chengyang District People’s Hospital , Qingdao , China
4 School of Life Sciences, Shandong University , Qingdao , China
Uversky Vladimir
Electronic publication date: 2024 Aug 30
Publication date: 2024
Volume: 12
Electronic Location ID: e18023
Received 2024 Jun 19; Accepted 2024 Aug 9
Copyright: © 2024 Wang et al.
Copyright year: 2024
Copyright holder: Wang et al.
License: This is an open access article distributed under the terms of the Creative Commons Attribution License, which permits unrestricted use, distribution, reproduction and adaptation in any medium and for any purpose provided that it is properly attributed. For attribution, the original author(s), title, publication source (PeerJ) and either DOI or URL of the article must be cited.
License URL: https://creativecommons.org/licenses/by/4.0/

Keywords: Plasmid, Hemorrhoid, Whole genome sequencing, Antimicrobial resistance

Funding: Foundation of Qingdao Key Health Discipline Development Fund ODZDZK-2022098 National Key Research and Development Program of China 2022YFE0199800 Key R&D Program of Shandong Province 2020CXGC011305 National Natural Science Foundation of China 82271658 This work was supported by the Foundation of Qingdao Key Health Discipline Development Fund under grand number ODZDZK-2022098, the National Key Research and Development Program of China under grant number 2022YFE0199800, the Key R&D Program of Shandong Province under grant number 2020CXGC011305, and the National Natural Science Foundation of China under grant number 82271658. The funders had no role in study design, data collection and analysis, decision to publish, or preparation of the manuscript.

==============================
Background

Hemorrhoids are common conditions at or around the anus, to which numerous people suffer worldwide. Previous research has suggested that microbes may play a role in the development of hemorrhoids, and the origins of these microbes have been preliminarily investigated. However, no detailed research on the microbes related to hemorrhoid patients has been conducted. This work aims to provide an initial investigation into the microbes related to hemorrhoid patients with high quality whole genome sequencing.

Methods

Forty-nine bacterial strains were isolated from seven hemorrhoid patients. Third-generation nanopore sequencing was performed to obtain high quality whole genome sequences. The presence of plasmids, particularly new plasmids, along with antibiotic resistance genes, was investigated for these strains. Phylogenetic analysis and genome comparisons were performed.

Results

Out of the 31 plasmids found in the strains, 15 new plasmids that have not been observed previously were discovered. Further structural analysis revealed new multidrug-resistant conjugative plasmids, virulent plasmids, and small, high-copy mobile plasmids that may play significant functional roles. These plasmids were found to harbor numerous integrases, transposases, and recombinases, suggesting their ability to quickly obtain genes to change functions. Analysis of antibiotic resistance genes revealed the presence of antibiotic resistant-integrons. Together with the surprising number of new plasmids identified, as well as the finding of transmission and modification events for plasmids in this work, we came to the suggestion that plasmids play a major role in genetic plasticity.

Conclusion

This study reveals that the diversity of plasmids in human-associated microbes has been underestimated. With the decreasing cost of whole-genome sequencing, monitoring plasmids deserves increased attention in future surveillance efforts.

Introduction

Hemorrhoids are vascular cushions located beneath the distal rectal mucosa, composed of vascular tissues, smooth muscles, and connective tissues (Stratta, Gallo & Trompetto, 2021). When this part of the normal anorectum becomes pathological, it develops into what is commonly known as hemorrhoids, resulting in swelling and painless rectal bleeding (Jacobs, 2014). Hemorrhoids are a frequently-occurring benign anal disease of the anorectal system that affects 4.4% to 39% of the general population, with the highest occurrence noted among individuals aged 45 to 65 (Johanson & Sonnenberg, 1990; Riss et al., 2012; Sun & Migaly, 2016). Based on the origin relative to the dentate line, hemorrhoids can be classified as internal, external, or mixed hemorrhoids (Jacobs, 2014). The primary and severe complications of hemorrhoids include perianal thrombosis and incarcerated prolapsed internal hemorrhoids with subsequent thrombosis (Wronski, 2012; Lohsiriwat, 2015). Hemorrhoidal disease increases the financial burden on health systems, including direct costs and missed workdays (Rubbini, Ascanelli & Fabbian, 2018). Sensitive symptoms such as anal bleeding, pain, and itchy sensations significantly impact patients’ quality of life, causing both physical and psychological distress (Riss et al., 2012; Kibret, Oumer & Moges, 2021).

The exact etiology of hemorrhoids remains unclear; however, it is most likely complex, involving the sliding anal cushion, hemorrhoid plexus hyperfusion, vascular anomalies, tissue inflammation, and internal rectal prolapse (rectal redundancy) (Lohsiriwat, 2012, 2015; Sandler & Peery, 2019). Studies have shown that the most frequently recognized risk factors for the development of hemorrhoids include inadequate dietary fiber, constipation, diarrhea, hypertension, high body mass index (BMI), pregnancy, and old age (Jacobs, 2014; Lee et al., 2014; Peery et al., 2015; Feyen et al., 2022). Recent investigations showed different signatures of bacterial community structures in hemorrhoids, suggesting a possible influence of bacteria in the development of hemorrhoids, and the presence of microbes in thrombosed hemorrhoids was identified (Wang et al., 2024). Despite these progresses, no investigations have been performed to study the microbes in hemorrhoid-suffering patients.

This work aims to address this issue by conducting whole-genome sequencing analysis of bacterial strains isolated from seven hemorrhoid patients. Surprisingly, a wide range of plasmid diversity was found, and the antibiotic resistance properties were discussed.

Materials and methods

Strain acquisition and genome preparation

The samples were taken from various regions of patients with hemorrhoids in the anorectal department of Qilu Hospital of Shandong University (Qingdao). After the patient signs the paper-based informed consent form, seven different sites (healthy buttock skin, anal skin, the outer side of the hemorrhoid, the inner side of the hemorrhoid, healthy anal gland, feces, and the hemorrhoid tissue) on the patient were swabbed with a sterile cotton swab and then immersed in the preservation solution. A total of seven hemorrhoid patients were recruited for the isolation of bacterial strains. After being transported to the laboratory at a low temperature, the solution was cultured on tryptic soy broth (TSB) medium. Dominant bacterial strain was selected for enrichment culture, and the genome was extracted using Tiangen Bacteria DNA Kit (Tiangen Biochemical Technology Co., Ltd., Beijing, China) and stored at −20 °C. Agarose gel electrophoresis is used to check the DNA quality. Ultraviolet spectrophotometry is used to assess the purity of DNA samples, while the fluorescent method is employed to determine the concentration of DNA samples.

Library construction and nanopore sequencing

The DNA library was prepared according to the rapid library construction method. The barcode kit SQK-RBK114.96 was used. The third-generation sequencer used was Nanopore P2solo with R10.4.1 flow cell. Basecalling was performed using Dorado version 0.5.3 (https://github.com/nanoporetech/dorado/) to obtain raw data. NanoQC version 0.9.4 and NanoPlot version 1.42.0 were used for data quality control, and Chopper version 0.8.0 was used to filter the data and generate clean data (De Coster & Rademakers, 2023). The Flye assembler version 2.8.1-b1676 was utilized to assemble the genome (Kolmogorov et al., 2019). This sequence was then corrected twice using Medaka version 1.12.0 (https://github.com/nanoporetech/medaka) to acquire the final whole-genome sequence for further analysis. Quast version 5.0.2, CheckM version 1.0.1, and BUSCO version 5.2.2 were used to evaluate the assembly quality, contamination, and integrity of the genome, respectively (Parks et al., 2015; Mikheenko et al., 2018; Manni et al., 2021). Ineligible samples were resequenced until higher quality assembly results were obtained for all samples. Genome assembly was performed with the Prokaryotic Genome Annotation Pipeline (Li et al., 2021). Genome and plasmid circularity were determined with Flye.

Bioinformatics

Taxonomy of all isolated strains was identified to the species level using GTDB-Tk version 2.1.1 (Chaumeil et al., 2022). Antibiotic resistance genes (ARGs) were annotated with AMRFinder version 3.11.26 (Feldgarden et al., 2021). MLST analysis of the genomes was performed with MLST version 2.23.0 (https://github.com/tseemann/mlst). BURST was used to group sequence types (Jolley, Bray & Maiden, 2018). The PubMLST database (https://pubmlst.org/) was utilized for sequence type analysis (Jolley & Maiden, 2010). Plasmids were typed with plasmidfinder version 2.1.1 (Carattoli & Hasman, 2020). Phylogenetic analysis was performed with SNP-based genome phylogenetic method using Snippy version 4.6.0 (https://github.com/tseemann/snippy). A maximum likelihood tree was constructed using FastTree version 2.1 (Price, Dehal & Arkin, 2010). The phylogenetic trees were visualized with the iTOL tool (Letunic & Bork, 2021). Plasmid maps were generated with SnapGene version 7.2.1. Single Nucleotide Polymorphism (SNP) calculations were performed using Snippy version 4.6.0.

Genomic data

The genomic data involved in this work can be found at GenBank under accession number PRJNA1119654.

Ethics

The study was conducted in accordance with the Declaration of Helsinki and approved by the Medical Ethics Committee of Qilu Hospital of Shandong University (Qingdao) (approval number KYLL-2023045, approval date Aug 11, 2023).

Results

Sample collection, bacteria isolation, and whole genome sequencing

Samples were taken from seven hemorrhoid patients who were subject to surgery in Qilu Hospital (Qingdao). Seven samples were taken from each patient, each from the healthy buttock skin, anal skin, the outer side of the hemorrhoid, the inner side of the hemorrhoid, healthy anal gland, feces, and the hemorrhoid tissue, totaling 49 samples. One bacterial strain was isolated from each sample by plating on TSB plates. Third-generation nanopore sequencing was performed with these strains, generating high quality whole genome sequences. Taxonomic classification was performed with whole genome sequences, leading to species-level classification.

An average of 3.3 contigs were assembled for each genome, including 44.4% (72 contigs) that are circular. Considering 32 of the strains contain plasmids, the low number of contigs for each genome and the high percentage of circular contigs suggest the good level of completeness of the genomes.

The 49 bacterial strains isolated include 34 Escherichia coli strains, six Klebsiella pneumoniae strains, four Enterococcus faecalis strains, two Klebsiella quasipneumoniae strains, two Enterobacter hormaechei strains, and one Morganella morganii strain (Table 1). The majority of the strains (89.8%) are Gram-negative bacteria belonging to a single order Enterobacterales. It is worth noting that all of the isolated strains are opportunistic pathogens commonly found with strong multidrug-resistant phenotypes. A maximum likelihood phylogenetic tree was constructed for all the E. coli strains isolated in this work (Fig. 1). The 34 strains were categorized into 13 sequence types, with ST10, ST1415, and ST4238 forming a single group (according to BURST analysis). In general, it can be observed that strains with the same sequence type are phylogenetically close, suggesting they are the same strain. The same strains are mostly from the same patient, except for two cases: patient Z22 (strains Z22-2, Z22-4, and Z22-5) and patient Z135 (all seven strains) share the same E. coli strain, whereas patient Z81 (Z81-7) and Z226 (Z226-7) share the same E. coli strain.

Table 1 Isolated strains and patient data.

Patient	Age	Gender	Strain	Isolated location	Taxonomy	
Z22	42	Male	Z22-1	Anal skin	E. coli	
Z22-2	The outer side of the hemorrhoid	E. coli	
Z22-3	Healthy buttock skin	M. morganii	
Z22-4	The inner side of the hemorrhoid	E. coli	
Z22-5	Healthy anal gland	E. coli	
Z22-6	Feces	E. coli	
Z22-7	The hemorrhoid tissue	E. coli	
Z81	35	Female	Z81-1	Anal skin	E. faecalis	
Z81-2	The outer side of the hemorrhoid	E. hormaechei	
Z81-3	Healthy buttock skin	E. coli	
Z81-4	The inner side of the hemorrhoid	E. coli	
Z81-5	Healthy anal gland	E. coli	
Z81-6	Feces	E. faecalis	
Z81-7	The hemorrhoid tissue	E. coli	
Z135	61	Male	Z135-1	Anal skin	E. coli	
Z135-2	The outer side of the hemorrhoid	E. coli	
Z135-3	Healthy buttock skin	E. coli	
Z135-4	The inner side of the hemorrhoid	E. coli	
Z135-5	Healthy anal gland	E. coli	
Z135-6	Feces	E. coli	
Z135-7	The hemorrhoid tissue	E. coli	
Z142	43	Male	Z142-1	Anal skin	E. coli	
Z142-2	The outer side of the hemorrhoid	E. coli	
Z142-3	Healthy buttock skin	E. coli	
Z142-4	The inner side of the hemorrhoid	E. coli	
Z142-5	Healthy anal gland	E. coli	
Z142-6	Feces	E. coli	
Z142-7	The hemorrhoid tissue	E. coli	
Z173	31	Female	Z173-1	Anal skin	K. pneumoniae	
Z173-2	The outer side of the hemorrhoid	K. quasipneumoniae	
Z173-3	Healthy buttock skin	K. pneumoniae	
Z173-4	The inner side of the hemorrhoid	K. quasipneumoniae	
Z173-5	Healthy anal gland	K. pneumoniae	
Z173-6	Feces	K. pneumoniae	
Z173-7	The hemorrhoid tissue	K. pneumoniae	
Z217	20	Male	Z217-1	Anal skin	E. faecalis	
Z217-2	The outer side of the hemorrhoid	E. coli	
Z217-3	Healthy buttock skin	E. faecalis	
Z217-4	The inner side of the hemorrhoid	E. coli	
Z217-5	Healthy anal gland	E. coli	
Z217-6	Feces	E. coli	
Z217-7	The hemorrhoid tissue	E. coli	
Z226	52	Female	Z226-1	Anal skin	E. coli	
Z226-2	The outer side of the hemorrhoid	K. pneumoniae	
Z226-3	Healthy buttock skin	E. hormaechei	
Z226-4	The inner side of the hemorrhoid	E. coli	
Z226-5	Healthy anal gland	E. coli	
Z226-6	Feces	E. coli	
Z226-7	The hemorrhoid tissue	E. coli	

Figure 1 Maximum likelihood phylogenetic tree constructed for all E. coli strains.

High resolution identification of plasmids from isolated strains

One particular strength of nanopore sequencing, or any 3rd generation sequencing, is the ability to generate long reads, thus making recovery and assembly of plasmid sequences more feasible than 2nd generation short-read high-throughput sequencing methods. The significantly improved single-strand accuracy of Q20+ chemistry, in combination with the R10.4.1 flow cell enables nanopore sequencing to generate long, high-quality (Q value > 18) reads (Zhang et al., 2023), which permits highly accurate discovery of plasmids. In this work, a total of 31 unique plasmids with an average size of 69.2 kb were found, of which 22 (71.0%) were assembled into circular form, indicating the completeness of plasmid sequences (Table 2). The high level of complete plasmid sequences identified suggests the power of Nanopore long-read high-throughput sequencing.

Table 2 Unique plasmids identified in this work.

Plasmid	Host strain	Contig	Circularity	Coverage	Size (kb)	Incompatibility group	MDR	New or old	Note	
pZ2221	Z22-2	Contig 4	Yes	9	5.7		No	New		
pZ2222	Z22-2	Contig 3	No	25	139.1	IncFIA/IncFIB/IncFII/Col156	Yes	Old	Same as pZ2241 and pZ2251	
pZ2223	Z22-2	Contig 5	No	32	16.4		No	Old		
pZ2224	Z22-2	Contig 8	No	25	7.2	IncFII	Yes	Old		
pZ2252	Z22-5	Contig 3	Yes	990	1.6		No	Old		
pZ2253	Z22-5	Contig 1	Yes	867	0.9		No	New		
pZ2261	Z22-6	Contig 2	Yes	197	4.1		No	Old	Same as pZ2271, K.pneumoniae origin	
pZ8111	Z81-1	Contig 2	Yes	24	6.9	rep9b	No	Old	Same as pZ8161	
pZ8112	Z81-1	Contig 3	No	96	3.2		No	Old	Same as pZ8162	
pZ8171	Z81-7	Contig 6	Yes	18	121.9	IncFIA/IncFIB/IncFII	Yes	New		
pZ8172	Z81-7	Contig 4	No	997	3.8		No	Old		
pZ13571	Z135-7	Contig 2	Yes	48	6.6		No	Old		
pZ14211	Z142-1	Contig 2	Yes	16	81.4	IncFIA/IncFIB	No	New		
pZ14212	Z142-1	Contig 3	Yes	82	4.1	IncN	Yes	Old		
pZ14221	Z142-2	Contig 2	Yes	27	94.5	IncFII	No	New	Same as pZ14241 and pZ14271	
pZ14231	Z142-3	Contig 2	No	15	156.0	IncHIB/IncU	Yes	New		
pZ14232	Z142-3	Contig 3	Yes	63	48.7	IncFIB	Yes	New		
pZ14251	Z142-5	Contig 2	Yes	20	101.8	IncY	No	New		
pZ14261	Z142-6	Contig 1	Yes	34	104.7	IncFIA/IncFIB	No	New		
pZ17311	Z173-1	Contig 2	Yes	15	203.0	IncFI	No	Old	Same as pZ17371	
pZ17321	Z173-2	Contig 4	Yes	14	172.2	IncFIB/IncHIB	No	New	Same as pZ17341	
pZ17322	Z173-2	Contig 2	No	750	8.1		No	New		
pZ17331	Z173-3	Contig 3	Yes	14	255.1	IncFIB	No	Old	Same as pZ17351	
pZ17361	Z173-6	Contig 1	No	11	141.6	IncFIB	No	Old		
pZ21751	Z217-5	Contig 3	Yes	15	128.2	IncFIB/IncFIC	Yes	New	Same as pZ21761 and pZ21771	
pZ22612	Z226-1	Contig 3	Yes	73	49.0	IncP1	No	Old		
pZ22613	Z226-1	Contig 2	Yes	201	11.7		No	New		
pZ22621	Z226-2	Contig 2	Yes	12	181.6	IncHI1B/IncFIB	No	Old		
pZ22651	Z226-5	Contig 2	Yes	22	57.5	IncFIB/IncFIC	No	New	Same as pZ22661	
pZ22662	Z226-6	Contig 1	No	12	90.2	IncI1	No	Old		
pZ22663	Z226-6	Contig 4	Yes	154	12.2		No	New		

This work identifies 15 new unique plasmids, nearly half of the total of 31 unique plasmids found, suggesting that there are numerous new plasmids in nature that have not been adequately investigated.

The structures of the 15 newly identified plasmids were subjected to more detailed analysis (Fig. 2). One initial strong impression is the high number of genes coding for integrase/transpose/recombinase. Take pZ14232, for example, out of the 56 protein-coding genes this plasmid has, 21 (37.5%) genes are integrase/transposase/recombinase-coding genes. This is true for all the large plasmids found in this work. Considering all new plasmids are from Enterobacteriales, this finding suggests the strong potential for plasmids of Enterobacteriales to load/unload cargo genes for modification of functions. This also suggests the plasticity of genomes of these opportunistic pathogens to varying environments during co-evolution with humans.

Figure 2 Structures of the 15 newly identified plasmids.

Different types of genetic elements are indicated by colored arrows, with red elements distinguished by black thin arrows

The sizes of the newly identified plasmids range from 0.9 to 172.2 kb. All plasmids, except for pZ2221, carry the rep gene essential for plasmid replication, supporting their annotation as plasmids. pZ2221 does not encode a Rep protein homologous to known Rep proteins. Considering the coverage of pZ2221 is only nine, we cannot rule out the possibility that it is part of a larger plasmid, and the circularity of this plasmid is simply an assembly error.

Four of the remaining 14 plasmids (pZ2253, pZ17322, pZ22613, and pZ22663) that carry at least one rep gene are relatively small in size and simple in structure. It is interesting to note that pZ2253, pZ22613, and pZ22663 carry only the rep gene and mobilization-related genes, suggesting they are small and mobile plasmids. In particular, their predicted copy numbers are respectively 45, six, and 13, respectively, indicating they have high copy numbers. This may imply they may play important unknown physiological roles in Enterobacteriales.

Three of the plasmids (pZ8171, pZ14261, and pZ21751) are conjugative plasmids that harbor the conjugative operons belonging to type IV secretion systems. Two of them (pZ8171 and Z21751) are multidrug-resistant plasmids, making them novel dangerous carriers of antimicrobial resistance that can disseminate within bacterial communities. Two additional plasmids, pZ14231 and pZ14232, are multidrug-resistant plasmids that do not encode type IV secretion systems but have mobility-related genes. They may also be involved in antimicrobial resistance dissemination.

These newly identified plasmids also carry other physiological functional gene clusters. Besides the widespread toxin-antitoxin systems that can be found on almost every large plasmid, pZ21751 has a colicin biosynthesis and immunity gene cluster making it a clear virulence factor. Additionally, pZ14211, one of the two plasmids in multidrug-resistant E. coli Z142-1, encodes a plasmid-borne type II secretion system. This protein complex is capable of exporting folded proteins, including toxins (Shaliutina-Loginova, Francetic & Doležal, 2023). The presence of these gene clusters suggests their role in virulence, host-microbe interactions, and microbe-microbe interactions. Metal resistance gene clusters were found on pZ14251 (mercury resistance cluster) and pZ14261 (copper/silver resistance cluster). Interestingly, pZ14231 carries a nitrate respiration gene cluster, whereas pZ14231 and pZ14251 carry cellulose biosynthesis gene clusters, granting the host bacteria additional feats.

Antibiotic resistance determinants

The presence of ARGs in each strain was analyzed with AMRFinder (Table 3). Twenty-five strains were found to contain at least one ARG. When not considering strains carrying ARGs that are part of the core genomes, acquired resistance mechanisms can be found in at least nine strains. Taking into account that the strains were isolated without supplementation of antibiotics in the media, the “lower-than-expected” acquired resistance prevalence (0% for Klebsiella strains, 29.4% for E. coli) suggests that the antibiotic resistance problem has not developed beyond rescue. It needs to be noted that, however, out of the seven multidrug-resistant plasmids identified, four are new plasmids. This reflects how little we know about multidrug-resistance plasmids, in consistence with the finding that only about half of the unique plasmids found in this work are previously identified.

Table 3 Antibiotic resistance genes in each strain.

Strain	Taxonomy	Chromosomal ARGs	Plasmid-born ARGs	
Z22-2	E. coli	–	dfrA17, aadA5, qacEΔ1, sul1, mphA, sul2, aph(3’’)-Ib, aph(6)-Id, tetA, blaCTX-M-27, floR, tetA, mphA, ermB	
Z22-3	M. morganii	blaDHA, sul1, qacEΔ1, aadA5, dfrA17, catA1, blaTEM-1, aac(3)-IId, tetB, sul1, qacEΔ1, aadA2, catA2	–	
Z22-4	E. coli	–	dfrA17, aadA5, qacEΔ1, sul1, mphA, sul2, aph(3’’)-Ib, aph(6)-Id, tetA, blaCTX-M-27,	
Z22-5	E. coli	–	dfrA17, aadA5, qacEΔ1, sul1, mphA, sul2, aph(3’’)-Ib, aph(6)-Id, tetA, blaCTX-M-27,	
Z81-1	E. faecalis	lsaA, dfrG	–	
Z81-2	E. hormaechei	oqxA, oqxB, fosA, blaACT-17	–	
Z81-6	E. faecalis	lsaA, dfrG	–	
Z81-7	E. coli	mphA, erm, aac(3)-IIe, dfrA17, aadA5	tetB, mphA, blaCTX-M-14	
Z142-1	E. coli	–	blaTEM-135, aph(3’’)-Ib, aph(6)-Id	
Z142-3	E. coli	–	tetA, qnrS, blaTEM-1, aac(3)-IId, tetA, floR, blaTEM-1, aadA1, qacL, dfrA14, qnrS1, tetA	
Z173-1	K. pneumoniae	blaSHV, fosA, oqxA, oqxB	–	
Z173-2	K. quasipneumoniae	fosA, oqxA, oqxB, blaOKP-B-1	–	
Z173-3	K. pneumoniae	blaSHV-144, fosA, oqxA, oqxB	–	
Z173-4	K. quasipneumoniae	fosA, oqxA, oqxB, blaOKP-B-1	–	
Z173-5	K. pneumoniae	blaSHV-144, fosA, oqxA, oqxB	–	
Z173-6	K. pneumoniae	blaSHV-11, fosA, oqxA, oqxB	–	
Z173-7	K. pneumoniae	blaSHV, fosA, oqxA, oqxB	–	
Z217-1	E. faecalis	lsaA	–	
Z217-3	E. faecalis	lsaA	–	
Z217-5	E. coli	blaCTX-M-182, sul2	aac(3)-IIe, blaTEM-1, sul2, aph(3”)-Ib, aph(6)-Id, tetA, floR, dfrA17, aadA5, qacEΔ1, sul1, mphA	
Z217-6	E. coli	blaCTX-M-182, sul2	aac(3)-IIe, blaTEM-1, sul2, aph(3”)-Ib, aph(6)-Id, tetA, floR, dfrA17, aadA5, qacEΔ1, sul1, mphA, ermB	
Z217-7	E. coli	blaCTX-M-182, sul2	aac(3)-IIe, blaTEM-1, sul2, aph(3”)-Ib, aph(6)-Id, tetA, floR, dfrA17, aadA5, qacEΔ1, sul1, mphA, ermB	
Z226-1	E. coli	–	bla CTX-M-27	
Z226-2	K. pneumoniae	blaSHV-11, fosA, oqxA, oqxB	–	
Z226-3	E. hormaechei	oqxA, oqxB, fosA, blaACT-16	–	

Class 1 integrons can be found in four strains (Fig. 3). Two integrons were found on the chromosome of M. morganii Z22-3, whereas one integron was found in pZ21751 in E. coli Z217-5, pZ21761 in E. coli Z217-6, and pZ21771 in E. coli Z217-7. The structures of these integrons have been previously observed. It is worth noting that the presence of integrons in M. morganii strains has not been widely reported, which could have been a rare event.

Figure 3 Structures of two integrons in this study.

Plasmid transmission and modification events

By comparing the strains found in this work, it is evident that plasmid transmission and modification events are quite common. Through SNP-based genomic phylogenetic analysis, we identified that E. coli Z22-2, Z22-4, Z22-5 strains are the same strain. They all carry a 139.1-kb multidrug resistance plasmid (pZ2222 in E. coli E22-2). However, E. coli Z22-2 carries three additional plasmids: pZ2221, pZ2223, and pZ2224, while E. coli Z22-5 carries an additional plasmid, pZ2253. Notably, pZ2224 is an IncFII-type multidrug resistance plasmid that can grant bacteria with additional functions. The variation in plasmid composition within the same strain from the same patient indicates the rapid spread of plasmids, leading to genetic adaptability.

Similarly, in patient Z22, E. coli strains Z22-1, Z22-6, Z22-7 are the same strain. However, strains Z22-6 and Z22-7 carry a plasmid that strain Z22-1 does not possess. In patient Z173, K. quasipneumoniae strains Z173-2 and Z173-4 are the same strain, but strain Z173-2 carries a pZ17322 plasmid that is not found in strain Z173-4; in patient Z226, E. coli strains Z226-5 and Z226-6 are the same strain, but strain Z226-6 carries two additional plasmids, pZ22662 and pZ22663, compared to strain Z226-5.

Plasmid modification events were also observed in this study. In patient Z217, E. coli strains Z217-5, Z217-6, and Z217-7 are the same strain, and the plasmids they harbor, pZ21751, pZ21761, and pZ21771, are the same plasmid. However, pZ21751 lost an ermB gene compared to pZ21761 and pZ21771, which could potentially make E. coli Z217-5 sensitive to erythromycin.

It needs to be noted here that very few SNPs were found between the chromosomes of the same strains. E. coli Z22-1 and Z22-6 have 28 and 0 SNPs in comparison with strain Z22-7. E. coli Z217-5 and Z217-6 have 175 and 141 SNPs in comparison with strain Z217-7. This is negligible (mutation rate < 0.01%) compared to genomic changes incurred with plasmid transmission and modification, which are frequently observed in this work. Therefore, we believe changes in plasmids are the primary source of genetic plasticity in the case of Enterobacteriales that are primarily studied in this work. Taking into account that a large portion of plasmids found in this work are new, we believe genetic plasticity has been underestimated.

Discussion

Currently, there is limited research on the association between hemorrhoids and microbiota. In our previous study, we investigated the microbial composition and characteristics at different sites in hemorrhoid patients based on 16S rRNA gene sequencing (Wang et al., 2024). Building upon this foundation, our current research delves deeper into the genomic information of microbiota associated with hemorrhoids, aiming to provide further insights for clinical treatment. In this study, it was found that different patients carried bacteria with highly similar genomes but different plasmids. Patient information indicated they were from different communities and underwent surgery in the same department 3 months apart. Despite a low nosocomial infection rate, this possibility cannot be ruled out. The resolution at which such conclusions can be made again suggests the feasibility of using whole genome sequences to track sources and transmission paths of pathogens.

Plasmids are essential bacterial genetic structures that play key roles in bacterial physiological versatility. They grant bacteria with additional physiological feats including antibiotic resistance, metal resistance, pathogenicity, and sometimes additional metabolic pathways. Acquisition of physiological features with plasmids is much faster than conventional evolution through mutation and selection. Therefore, the content of plasmids has always been a central piece when comparing different strains of the same species. However, 2nd generation sequencing struggles with assembling plasmids, which hinders the discovery of new plasmids. This study utilizes 3nd generation nanopore sequencing technology to obtain high-quality plasmid information, effectively addressing the shortcomings of 2nd generation sequencing. It identified 15 new plasmids among 49 strains of bacteria in their natural state, achieving a surprisingly high detection rate of new plasmids. As an essential genetic element in bacteria, plasmids frequently undergo transfer and mutation. They often harbor antibiotic resistance genes, which can result in treatment failures in clinical settings. In a single patient, bacteria may migrate and settle along with feces or other physiological processes. Plasmids, which travel alongside bacteria, can transfer between strains via mechanisms like conjugation and transformation. When external pressures such as antibiotic treatment are present, plasmids containing resistance genes tend to persist. Mutations linked to antibiotic resistance that arise during replication are also prone to selection and retention. This study also noted bacterial strains from the same patient, located in distinct sites, possessing identical genomes but hosting diverse plasmids. Analyzing these plasmids genetically could greatly advance our comprehension of bacterial transmission routes in diseases, providing critical insights for clinical treatment approaches.

Although we selected multiple patients and sampled genomes of bacteria associated with hemorrhoids at various sites, there are still significant limitations. Due to cost constraints, we did not sequence every isolated bacterium from each sample, leading to some loss of information. Nevertheless, we identified several new plasmids, highlighting the significance of third-generation sequencing in directly tracking plasmids.

Conclusions

Third-generation nanopore sequencing was used to analyze 49 bacterial strains from hemorrhoid patients to study the properties of strains isolated from these patients. With high-quality whole-genome sequences, we identified 15 new plasmids, indicating a high level of plasmid diversity not previously observed. Further analysis confirmed plasmid transmission and modification events occurring within the same individual. Although antibiotic resistance gene carriage was not prevalent, antibiotic resistance integrons were detected, highlighting genetic plasticity. This study reveals that we have underestimated the diversity of plasmids in human-carrying microbes. As whole-genome sequencing becomes more affordable, monitoring plasmids deserves attention in the future.

Supplemental Information

Supplemental Information 1 STROBE Statement.

Supplemental Information 2 Genomic sequences for subject Z22.

Supplemental Information 3 Genomic sequences for subject Z81.

Supplemental Information 4 Genomic sequences for subject Z135.

Supplemental Information 5 Genomic sequences for subject Z217.

Supplemental Information 6 Genomic sequences for subject Z142.

Supplemental Information 7 Genomic sequences for subject Z173.

Supplemental Information 8 Genomic sequences for subject Z226.

Additional Information and Declarations

Competing Interests

Author Contributions

Human Ethics

Data Availability

The authors declare that they have no competing interests.

Yihua Wang performed the experiments, prepared figures and/or tables, and approved the final draft.

Wenya Su performed the experiments, prepared figures and/or tables, and approved the final draft.

Xiang Zeng performed the experiments, authored or reviewed drafts of the article, and approved the final draft.

Zhaopeng Liu performed the experiments, authored or reviewed drafts of the article, and approved the final draft.

Jiaming Zhu performed the experiments, authored or reviewed drafts of the article, and approved the final draft.

Mingyu Wang conceived and designed the experiments, analyzed the data, authored or reviewed drafts of the article, and approved the final draft.

Ling Li conceived and designed the experiments, authored or reviewed drafts of the article, and approved the final draft.

Wenlong Shen conceived and designed the experiments, analyzed the data, authored or reviewed drafts of the article, and approved the final draft.

The following information was supplied relating to ethical approvals (i.e., approving body and any reference numbers):

The Medical Ethics Committee of Qilu Hospital of Shandong University (Qingdao)

The following information was supplied regarding data availability:

The genomic data involved in this work is available at Genbank: PRJNA1119654.

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
