# Peer review of "Surprising diversity of new plasmids in bacteria isolated from hemorrhoid patients"

_PeerJ, doi:10.7717/peerj.18023_

## Round 0.1 · original submission · Minor Revisions

Please address concerns of both reviewers and amend manuscript accordingly.

·

Basic reporting

In some text details, there were spelling mistakes and grammatical errors. Please correct them.

In the abstract, there was no conclusion. Please conclude your results and discussion. Do you follow the abstract structure as usual?

Be consistent, in manuscript you wrote the Results and Discussion, also conclusion; but in abstract you mentioned only Results without conclusion.

Experimental design

Please explain more clearly the aim of this study and highlight the background. There are still unclear knowledge gap.

How did you find the patients and confirm the hemorrhoid?
Why did you choose the sample site, for example skin, hemorrhoid tissue or feses? Please explain the reason
In sample collection, how did you choose one bacteria strain?

Validity of the findings

The title says "Surprising diversity of new plasmids, it should be highlighted in discussion; what is the new plasmids that make surprise? The benefit of detecting this new plasmids can influence antibiotic resistance, please explain the pathogenesis.

In this study, you find the same strain on different hosts, how this strain could be transmitted to different people?

In line 146, do you mean ‘worth nothing’ or ‘worth noting’ ? It has different meaning.

There was no section for discussion, you put in with the results. It’s better to separate between results and discussion.
Discussion begins in line 171?

In the discussion you didn’t mention the plasmid transmission and modification events can happen in the same individual. How can this happen?

Please describe more in the pathogenesis of genetic plasticity can affect different plasmids in the same patient.

In conclusion, you wrote ‘surveillance of the plasmid will be a worthy attention in the future’. Please put in discussion before you can conclude this statement.

The conclusion should not repeat sentences from results. Make it concise, and write down the summary from the discussion.

Additional comments

This manuscript should be corrected in basic reporting, design or method. Add value in discussion and clinical application of this study.

Reviewer 2 ·

Basic reporting

The manuscript is written in clear and professional English. However, there are minor grammatical errors and awkward phrasings that could benefit from additional editing. For example, lines 51-52: "When this part of the normal anorectum becomes pathological, it develops into what is commonly known as hemorrhoids, resulting in swelling and painless recal bleeding." The term "recal bleeding" should be corrected to "rectal bleeding." The manuscript provides an adequate background on the topic of hemorrhoids and the role of microbes. The references cited are relevant and current. However, it would be beneficial to include more recent studies on the role of microbiota in hemorrhoid development to provide a broader context. The structure of the article is well-organized, following the standard format. Figures and tables are relevant, high-quality, well-labeled, and described. The raw data has been shared as per PeerJ policy, enhancing the manuscript's transparency. The manuscript presents a self-contained study with relevant results that address the initial hypotheses. The findings are logically presented and discussed.

Experimental design

The study is an original primary research project that fits well within the aims and scope of PeerJ. It addresses a novel area by exploring the diversity of plasmids in bacteria isolated from hemorrhoid patients.The research question is well-defined, relevant, and meaningful. The study aims to fill a knowledge gap by investigating the microbial diversity associated with hemorrhoid patients using whole genome sequencing.The investigation is conducted rigorously with high technical standards. Ethical approval has been obtained, and all necessary ethical guidelines have been followed.The methods section is detailed and provides sufficient information for replication. The description of the DNA sequencing and analysis techniques is thorough.

Validity of the findings

The impact and novelty of the findings are significant. The discovery of new plasmids and their potential roles in genetic plasticity are noteworthy contributions to the field. Replication is encouraged, as it would benefit the broader scientific community.The underlying data is robust, statistically sound, and well-controlled. The use of third-generation Nanopore sequencing provides high-quality whole genome sequences, supporting the validity of the findings.The conclusions are well-stated, directly linked to the original research question, and supported by the results. The authors effectively summarize the implications of their findings and suggest directions for future research.

Additional comments

The manuscript could benefit from a more thorough discussion of the limitations of the study and potential confounding factors. It would be helpful to include a brief discussion on the clinical implications of the findings, particularly regarding the management of hemorrhoid patients and the potential impact of microbial diversity on treatment outcomes.

---

## Round 0.2 · accepted · Accept

All concerns of the reviewers were addressed, and the revised manuscript is acceptable now.